# Prognostic Value of the Lung Immune Prognosis Index Score for Patients Treated with Immune Checkpoint Inhibitors for Advanced or Metastatic Urinary Tract Carcinoma

**DOI:** 10.3390/cancers15041066

**Published:** 2023-02-07

**Authors:** Pauline Parent, Edouard Auclin, Anna Patrikidou, Laura Mezquita, Nieves Martínez Chanzá, Clément Dumont, Alejo Rodriguez-Vida, Casilda Llacer, Rebeca Lozano, Raffaele Ratta, Axel S. Merseburger, Cora N. Sternberg, Giulia Baciarello, Emeline Colomba, Alina Fuerea, Benjamin Besse, Yohann Loriot, Pernelle Lavaud

**Affiliations:** 1Department of Cancer Medicine, Gustave Roussy, Université Paris-Saclay, 114 Rue Edouard Vaillant, 94800 Villejuif, France; 2Department of Oncology Medical, CHU Lille, 2 Av. Oscar Lambret, 59000 Lille, France; 3Medical Oncology Department, CHU Lille, University of Lille, 42 Rue Paul Duez, 59000 Lille, France; 4Department of Medical and Thoracic Oncology, Hôpital Européen Georges Pompidou, 20 Rue Leblanc, AP-HP, 75015 Paris, France; 5SIRIC CARPEM Comprehensive Cancer Center, University of Paris, 75006 Paris, France; 6Medical Oncology Department, Hospital Clinic of Barcelona, Laboratory of Translational Genomics and Targeted Therapies in Solid Tumors, IDIBAPS, 08036 Barcelona, Spain; 7Department of Medical Oncology, Jules Bordet Institute, 1070 Brussels, Belgium; 8Saint-Louis Hospital, AP-HP, 75010 Paris, France; 9Medical Oncology Department, Saint-Louis Hospital, Université Paris Cité, 75006 Paris, France; 10Hospital del Mar, 08003 Barcelona, Spain; 11Hospitales Universitarios Regional y Virgen de la Victoria de Málaga, 29010 Málaga, Spain; 12Spanish National Cancer Research Centre (CNIO), 28029 Madrid, Spain; 13Hospital Universitario de Salamanca, 37007 Salamanca, Spain; 14Medical Oncology Department, Foch Hospital, 92150 Suresnes, France; 15Department of Urology, University Clinic Schleswig-Holstein-Lübeck, 23562 Lübeck, Germany; 16Department of Medical Oncology, Weill Cornell Medicine, Meyer Cancer Center, Englander Institute for Precision Medicine, New York, NY 10021, USA; 17Department of Medical Oncology, Unit Fondazione IRCCS, Istituto Nazionale dei Tumori, 20133 Milan, Italy

**Keywords:** LIPI score, prognosis, urothelial cancer, immune checkpoint inhibitors, biomarker

## Abstract

**Simple Summary:**

In this report, we studied the role of the LIPI score, a biological score based on 2 factors, derived neutrophils/(leukocytes minus neutrophils) ratio and lactate dehydrogenase, in population with unresectable urothelial cancer treated with immune checkpoint inhibitor (ICI). This score was associated with clinical outcomes for ICI in several tumor types. In total, 137 and 541 patients were respectively enrolled in a retrospective ICI cohort and a validation cohort. LIPI classified the population of these cohorts in good (52–56%), intermediate (35–36%) and poor (9–12%) prognosis groups. Poor LIPI was associated with a poorer OS for the 2 cohorts. In patients with good prognosis according to the Bellmunt score, LIPI identifies a subset of patients with poorer outcomes. To conclude the LIPI score was associated with survival in unresectable urothelial cancer patients treated by ICI. Future prospective studies will be required to test the combination of Bellmunt score and LIPI score as a more accurate prognosis tool.

**Abstract:**

Few prognostic factors have been identified in patients with metastatic urothelial carcinoma (mUC) treated with immune checkpoint inhibitors (ICIs). The Lung Immune Prognostic Index (LIPI) was associated with clinical outcomes for ICIs in several tumor types. We aim to assess the value of the LIPI in patients with mUC treated with ICIs. A retrospective ICI cohort and a validation cohort (SAUL cohort) included, respectively, patients with mUC treated with ICI in 8 European centers (any line) and patients treated with atezolizumab in a second or further line. A chemotherapy-only cohort was also analyzed. The LIPI score was based on 2 factors, derived neutrophils/(leukocytes minus neutrophils) ratio (dNLR) > 3 and lactate dehydrogenase > upper limit of normal, and defined 3 prognostic groups. The association of LIPI with progression-free survival (PFS) and overall survival (OS) was assessed. In the ICI and SAUL cohorts, 137 and 541 patients were respectively analyzed. In the ICI cohort, mPFS and mOS were 3.6 mo (95% CI; 2.6–6.0) and 13.8 mo (95% CI; 11.5–23.2) whereas in the SAUL cohort the mPFS and mOS were 2.2 mo (95% CI; 2.1–2.3) and 8.7 mo (95% CI; 7.8–9.9) respectively. The LIPI classified the population of these cohorts in good (56%; 52%), intermediate (35%; 36%) and poor (9%; 12%) prognostic groups (values for the ICI and SAUL cohorts respectively). Poor LIPI was associated with a poorer OS in both cohorts: hazard ratio (HR) for the ICI cohort = 2.69 (95% CI; 1.24–5.84, *p* = 0.035); HR = 2. 89 for the SAUL cohort (CI 95%: 1.93–4.32, *p* < 0.0001). Similar results were found in the chemo cohort. The LIPI score allows to identify different subgroups in patients with good prognostis according to the Bellmunt score criteria, with a subset of patients with poorer outcomes having an mOS of 3.7 mo compared to the good and intermediate LIPI subgroups with mOS of 17.9 and 7.4 mo, respectively. The LIPI score was associated with survival in mUC patients treated by ICIs. Future prospective studies will be required to test the combination of Bellmunt score and the LIPI score as a more accurate prognosis tool.

## 1. Introduction

Bladder cancer is the most common cancer of the urinary tract, with an estimated 575,000 newly diagnosed cases worldwide in 2020 [1]. Platinum-based chemotherapy has been the mainstay of treatment in the metastatic setting, but outcomes are poor, with an median overall survival (OS) of 12 to 15 months in patients eligible for first-line cisplatin-containing chemotherapy and around 9 months with patients “unfit” for cisplatin [2,3,4]. Immune checkpoint inhibitors (ICIs) have been approved for the management of platinum-pretreated metastatic urothelial cancer (mUC) and for cisplatin-ineligible patients with previously untreated mUC and a PD-L1 positive status [5,6,7,8,9,10]. More recently, it was found that maintenance treatment with avelumab after first-line platinum-based chemotherapy improved overall survival (OS) in patients whose disease had not progressed [11]. However, only a subset of patients benefit from ICIs, highlighting the need of more accurate selection of patients who are more likely to respond: in early pivotal trials in second-line and beyond therapy, the overall response rates (ORR) waere assessed between 16% and 21% [5,6,7,8].

The anticancer immune response that leads to the effective destruction of cancer cells depends on the immune tumor microenvironment. Systemic chronic inflammation is known to induce alterations in this microenvironment and is involved in the emergence of immune resistance in cancer [12,13]. As key mediators of the systemic inflammatory process, neutrophils suppress T cell proliferation and activation and may be regulated by cancer cells [14,15]. In addition, preclinical studies have indicated that systemic mobilization of neutrophils facilitates metastatic spread [16,17,18]. For this reason, leukocyte and neutrophil counts have been used to develop clinical indicators of systemic inflammation [19]. The derived neutrophil to leukocyte ratio (dNLR) has been proposed as a potentially more relevant ratio, as it includes not only lymphocytes, but also monocytes and eosinophils, thus better reflecting the host immune status. In addition, dNLR has been used as an NLR alternative in several clinical trials and independent significant association between high dNLR and poor OS as well as DFS outcomes was described in patients with malignancies [20]. Another metabolic and proliferation biomarker, lactate dehydrogenase (LDH), has been used to identify patients with poor prognosis: high LDH serum levels often reflect tumor burden and are present in aggressive forms of solid tumors [21]. In order to integrate these clinical indicators of systemic inflammation, the lung immune prognostic index (LIPI) score was developed as an accessible tool to estimate initially the clinical outcomes of lung cancer patients on ICIs treatment [22,23,24,25].

This score, based on dNLR > 3 and LDH greater than upper limit of normal (ULN), characterizes 3 groups (good, 0 factor; intermediate, 1 factor; poor, 2 factors). In non-small cell lung cancer patients, a low LIPI score was independently associated with overall survival (OS), progression-free survival (PFS) and disease control rate (DCR) with ICIs [22]. In a pooled validation analysis including data from 11 randomized trials, the LIPI score was confirmed as an independent prognostic factor irrespective of treatment modality in a cohort of 4914 pts treated with ICIs, targeted therapy or chemotherapy [24].

Few prognostic and predictive factors have been validated for ICIs treatment so far. The prognostic value of the LIPI in advanced/metastatic urinary carcinoma in ICI-treated patients is unknown. The aim of this study was to identify the LIPI as a prognosis score in two independent cohorts of patients with advanced or metastatic urothelial cancer treated with ICI and in a cohort of patients with mUC receiving only chemotherapy.

## 2. Patients and Methods 

### 2.1. Study Design and Patient Cohorts

Three cohorts of patients with mUC were retrospectively analyzed. The first cohort (“ICI cohort”) included patients treated with ICIs in everyday practice, expanded access, compassionate-use programs and clinical trials in eight high-volume cancer centers in Europe (from France, Spain and Belgium) between May 2013 and July 2019. Inclusion criteria were patients treated from January 2015 to January 2019 for an mUC or unresecable urothelial carcinoma, in first line treatment or more (immunotherapy combinations were allowed but not combinations of immunotherapy plus chemotherapy). The second cohort (“SAUL cohort”) was based on patients enrolled in the SAUL study (Clinicaltrials.gov NCT02928406). SAUL was a large single-arm study including 1004 patients which evaluated atezolizumab, a programmed death-ligand 1 (PD-L1) inhibitor, as a treatment for mUC in the real-world context between November 2016 and March 2018. The design and results of this study have been described in detail previously [26]. Briefly, patients with mUC, including populations typically excluded from clinical trials, received atezolizumab 1200 mg intravenously every 3 weeks until loss of clinical benefit or unacceptable toxicity was observed. The primary endpoint was safety; patients received a median of 5 cycles (range 1–28), with a median treatment duration of 2.8 months (mo) (range 0–19). Adverse events (AEs) (any grade) occurred in 88% patients, the most common being fatigue and decreased appetite. Median OS was 8.7 mo (95% CI, 7.8–9.9). Finally, the third cohort (“Chemo cohort”) included patients treated only with chemotherapy in four European cancer centers between September 2011 and July 2019.

This cohort explored the prognostic value of LIPI score in the setting of chemotherapy. 

Baseline demographic, clinical, pathological and biological data were collected for each patient. Radiological response was evaluated, and radiological criteria were noted: investigator, response evaluation criteria in solid tumors (RECIST) or modified RECIST (mRECIST). 

The LIPI score combines two variables: pretreatment LDH level, and pretreatment dNLR derived neutrophils/(leukocytes minus neutrophils) ratio. These defined 3 prognosis groups: good prognosis (LDH ≤ ULN and dNLR ≤ 3), intermediate prognosis (LDH > ULN or dNLR > 3) and poor prognosis (LDH > ULN and dNLR > 3) [22]. Bellmunt risk factors, i.e., ECOG performance-status score >0, Hb concentration < 10 g/dL, and presence of liver metastases were identified to calculate a Bellmunt score in the SAUL cohort [27]. 

### 2.2. Statistical Analysis

Mean (SD) values and frequencies were provided for the description of continuous and categorical variables, respectively. Means and proportions were compared using Student’s *t*-test and chi2-test (or Fisher’s exact test when number of patients < 20, if appropriate), respectively. PFS was defined as the time between ICI start and progression, or death, whichever occurred first. OS was defined as the time between ICI start and death from any cause. Patients without an endpoint event at the time of data cut-off were censored. DCR was defined as any best objective response other than progressive disease, and overall response rate (ORR) as the sum of complete and partial responses. Patients with fast disease progression were defined as patients who died within the first three months of immunotherapy. OS and PFS were estimated using the Kaplan-Meier method and described using median with 95% confidence intervals (95% CI). Follow-up was calculated using a reverse Kaplan–Meier estimation. Factors associated with endpoints were assessed with univariate Cox-proportional-hazard models, providing the hazard ratio (HR) and 95% CI. Variables were considered statistically associated with the endpoints if the univariate *p*-value of the Cox model was <0.10. Statistically significant and clinically relevant variables were included in the multivariate Cox models. In the SAUL cohort, the discrimination ability of the LIPI and Bellmunt scores was assessed by Harrell’s concordance index (c-index) and compared [28]. All analyses were performed using R Studio software (https://www.rstudio.com accessed on 1 January 2021). Values of *p* < 0.05 were considered statistically significant and all tests were two-sided.

The study was approved by the Institutional Review Board (IRB) of Gustave Roussy (N°2020-01) and by the IRBs of all participating European centers.

## 3. Results

### 3.1. Study Cohorts

The ICI cohort included 153 patients with mUC treated with ICIs between May 2013 and July 2019. Overall, 137 patients were analyzed (Figure 1). Median follow-up was 24 mo (95% CI; 19.8–30.4). In this cohort, 77 (56%), 48 (35%) and 12 (9%) patients had a good, intermediate and poor LIPI score, respectively. 

The SAUL study enrolled 1004 patients with disease progression after one to three prior therapies between November 2016 and March 2018. Owing to missing data, only 541 patients were analyzed (Figure 1). Median follow-up was 12.7 mo (95% CI; 11.8–13.2). Of 541 patients analyzed, 281 (52%), 198 (36%) and 63 (12%) patients had respectively a good, intermediate and poor LIPI score. 

Finally, the Chemo cohort included 85 patients with mUC treated exclusively with chemotherapy from September 2011 to July 2019. Overall, 67 patients were analyzed (Figure 1). Median follow-up was 39.1 mo (95% CI; 33.7-NR). Of the analyzed patients, 39 (58%), 21 (31%) and 7 (11%) patients had a good, intermediate and poor LIPI score, respectively. 

Baseline patient and disease characteristics are summarized in Table 1. Circulating inflammatory marker status as reflected by white blood cell (WBC) count, absolute neutrophil count (ANC), albumin, hemoglobin, and LDH level was similar between the three cohorts (Appendix A). 

### 3.2. Prognostic Value of LIPI Score in the ICI and SAUL Cohorts

In the ICI cohort, median (m) OS was 14.4 mo (95% CI; 10.8–20.2) and mPFS was 3.7 mo (95% CI, 2.69–5.91). Median OS were 19.7 mo (95% CI, 13.0–40.7), 13.7 mo (95% CI, 6.93–25.8) and 5.4 mo (95% CI 2.50-NR) for good, intermediate and poor LIPI groups, respectively (*p* = 0.002) (Figure 2). In a multivariate analysis integrating other known prognostic factors (performance status, liver metastatic site, hemoglobin and albumin), poor LIPI score retained a significant association with shorter OS (HR = 2.69, 95% CI; 1.24–5.84, *p* = 0.035) and shorter PFS (HR = 4.36, 95% CI; 2.04–9.33, *p* < 0.001). An intermediate LIPI score was correlated with a shorter PFS (HR = 1.68, 95% CI; 1.08–2.61, *p* < 0.001) (Table 2 and Appendix A). 

These results were confirmed in the SAUL cohort. Median OS was 8.7 mo (95% CI, 7.8–9.9) and mPFS was 2.2 mo (95% CI, 2.1–2.3). The good, intermediate and poor LIPI groups had a mOS of 12.4 (95% CI 10–NR), 5.4 (95% CI 4.5–6.9) and 2.4 mo (95% CI 1.61–3.7) respectively (*p* < 0.0001) (Figure 2). Patients with a poor and intermediate LIPI score had shorter OS, with a HR = 2.89 (95% CI; 1.93–4.32) and HR = 1.78; (95% CI; 1.35–2.33), respectively (*p* < 0.0001). Results were similar for PFS (*p* < 0.001) (Table 2 and Appendix A). 

### 3.3. Prognostic Value of LIPI Score in Chemo Cohort

We next investigated whether the LIPI was also prognostic for patients receiving chemotherapy. Median OS was 8.5 mo (95% CI, 6.3–11.3) and mPFS was 6.1 mo (95% CI, 4.7–7.1). The LIPI score was also associated with both PFS and OS (Figure 3). Patients with a good LIPI score had longer survival than patients with an intermediate and poor score, with an estimated median survival of 10.4 vs 6.0 vs 4.5 mo, respectively (*p* = 0.03) (Table 2 and Appendix A). 

### 3.4. Comparison of LIPI and Bellmunt Scores in Patients Receiving ICI sin the SAUL Cohort

We compared the prognostic value of the LIPI score and the Bellmunt score, a well-known prognostic factor form UC patients treated with platinum-based chemotherapy [27]. In the SAUL cohort, the Bellmunt score classified patients into 4 groups with different prognoses: 227 patients had a good prognosis and 13 patients a poor prognosis when LIPI and Bellmunt scores were combined (Appendix A). In the multivariate Cox analysis, a high Bellmunt score was associated with shorter OS and PFS (Table 3). Both LIPI and Bellmunt scores were independent prognostic factors for OS and PFS (Table 3). Sixty-six percent of patients with a high Bellmunt or LIPI score were patients with fast progressions (Table 3). The c-index was 0.66 (95% CI 0.60–0.71) and 0.67 (95% CI 0.59–0.74) for the LIPI score and Bellmunt score, respectively. When the LIPI and Bellmunt scores were combined in the same multivariable model, the c-index was 0.70 (95% CI: 0.65–0.76). This increase of the “combined” c-index was not significant (difference with the c-index of the Bellmunt score alone: −0.03, 95% CI; −0.08–0.01). 

To explore whether the LIPI score provides any additional value to the Bellmunt score, we assessed the LIPI score in patients with good prognosis according to the Bellmunt score (score 0–1). The LIPI score identified patients with poor prognosis (Figure 4). Median OS was 17.9 mo (95% CI; 12.7–NR), 7.4 mo (95% CI; 6.1–12.8) and 3.7 mo (95% CI; 3.5–NR) for good, intermediate and poor LIPI groups, respectively (*p* < 0.0001). Median PFS was 6.0 (IC 95%; 4.2–6.2), 2.5 (IC 95%; 2.2–4.0) and 2.5 (IC 95%; 2.0–7.2) mo (*p* = 0.004) for good, intermediate and poor LIPI groups, respectively (Table 3).

The LIPI and Bellmunt scores were equally prognostic in the following subgroups: bladder or upper tract cancer and metastatic or locally advanced cancer (Appendix A).

## 4. Discussion

In this analysis, we showed that patients treated with ICIs with a low LIPI score had better clinical outcomes than those with a high LIPI score. Our study includes utilization of a large cohort of patients treated outside of clinical trials in several centres in Europe and external validation in a cohort of 541 patients enrolled in a prospective study, thus ensuring a large overall cohort sample size. The prognostic role of the LIPI score observed in ICI-treated patients with mUC is consistent with that observed in other tumour types and supports the strong prognostic role of the LIPI in patients treated with ICIs [22,23,24,25]. Importantly, we showed that the LIPI score predicts the outcome of patients treated with chemotherapy despite a main limitation of this result being the small number of patients which thus had no predictive value to identify patients who are more or less likely to respond to ICIs. 

Currently, no fully validated predictive biomarker is available to predict the prognosis and antitumor efficacy of ICIs in patients with mUC. Programmed death ligand 1 expression, defective mismatch repair phenotype, and tumor mutational load have been studied as predictive biomarkers for the management of urothelial cancer, but none are currently used in daily practice, mainly due to the lack of reliable reproducibility [29].

As an example, the lack of standardization of PD-L1 assessment across immunohistochemical assays, plus its dynamic expression make interpretation of the results difficult [29]. TMB seems to be correlated with treatment responses to ICIs in different cancers and was investigated in mUC. The exploratory analyses of the phase II IMvigor210 trial showed that TMB was significantly higher in patients who responded to atezolizumab than in non-responders, however, the TMB was not able to perfectly predict the patients survival benefit [10]. 

Other biomarkers are emerging, such as circulating tumor DNA and microbiota, but these need to be validated in prospective clinical trials [30,31]. High ctDNA levels have been shown to be correlate with more aggressive forms of disease and to be conversely correlated with overall survival in patients treated with durvalumab [32]. In the IMvigor010 trial, patients with positive ctDNA had improved rates of disease-free survival and overall survival with adjuvant atezolizumab compared with patients treated with a placebo, while no difference was reported between the treatment arms for ctDNA-negative patients. Nevertheless, ctDNA level assessment is not implemented in clinical practice and the optimal predictive cut-off needs to be confirmed [33]. In this sense, the standardized easy-to-use LIPI scoring algorithm, alone or combined with other biomarkers could erase their intrinsic weaknesses and better assess the probability of response to ICIs in mUC; this would help avoid the exposure of patients to possible side effects when a minimal likelihood of a response is assumed. While the LIPI score is prognostic and not predictive, it may still be relevant as a tool to indirectly inform treatment discussions in mUC, especially given the context of potential access to emerging drugs such as FGFR3 inhibitors or antibody drug conjugates in the near future [34,35,36]. However, the LIPI score should first be tested prospectively in the clinical trials investigating these new drugs as the predictive role of the LIPI score is unknown. 

The LIPI score was based on LDH levels and pretreatment dNLR, and it reflects the systemic inflammatory status in patients with metastatic cancer. Several other inflammatory indices have been developed and studied in patients with genitourinary cancer. Recent meta-analyses have reported the prognostic values of such indices based on pretreatment NLR, and LDH in patients for UC patients at different disease stages [37,38,39]. A recent study identified NLR > 3 to be associated with a decreased response to neoadjuvant chemotherapy and shorter disease-specific survival and OS in nonmetastatic muscle-invasive bladder cancer [40]. In a recent multicenter retrospective study on 463 pembrolizumab-treated patients, a prognostic model based on ECOG PS, site of metastasis, hemoglobin levels, and NLR was developed and internally validated [41]. A five-factor prognostic model, based on pretreatment ECOG PS, presence of liver metastases, platelet count, NLR and LDH, has been validated within early stages clinical trials on 405 chemotherapy-pretreated mUC patients treated with PD-L1 inhibitors [42]. Another risk score was developed for advanced urothelial carcinoma treated with first-line ICIs. The score assigned ECOG PS ≥ 2, albumin < 3.5 g/dL, NLR > 5, and liver metastases each one point, with a higher score being associated with worse OS [43]. Overall, these data are concordant with the LIPI score that identified systemic inflammatory status as negative prognostic factors in mUC.

Historically, the Bellmunt score has been used as a stratification factor in clinical trials for patients treated with chemotherapy after progression with first-line platinum-based chemotherapy. In our study, the c-index was 0.66 and 0.67 for the LIPI score and Bellmunt score, respectively. When the LIPI and Bellmunt scores were combined in the same multivariable model, the c-index was 0.70. Even though the LIPI score did not outperform the Bellmunt score in our study, the results of our analysis indicate that the LIPI might be a good predictor of OS and PFS in advanced mUC patients with good prognosis according to the Bellmunt score. In addition, the LIPI includes two objective parameters whereas the Bellmunt score includes the subjective assessment of performance status [44]. A prospective study with 3621 patients treated for hematological malignancies found that a patient’s PS assessment was consistent with that of the physician in only 65% of cases [45].

Integrating parameters reflecting the biology of the tumor may improve the performance of such a score. A three-factor model including genomic (a single-nucleotide variant count > 9) and clinical (NLR < 5 and lack of visceral metastasis) variables was associated with benefit from ICIs but not from taxane therapy in 62 patients with metastatic UC and might be more accurate in predicting the prognosis as the Harrell’s C-statistic was 0.9 [46]. Several biomarkers of the ICIs response are being investigated, including PD-L1 protein expression, tumor mutational burden, T cell infiltration transcriptomic subtypes, and T cell pathway activation [47], but these have not as yet been shown to accurately identify patients who are more likely to benefit from ICIs. More comprehensive studies are needed to incorporate molecular biomarker data in the management of patients in daily practice.

We acknowledge several limitations of our study, including the retrospective design responsible for some imbalances in baseline patient characteristics and differences in follow-up, and its inherent selection bias and missing data. Recent data suggest also that mRECIST shows more ORR than RECIST 1.1, which could lead to variability according to the radiological criteria used [48].

However, our study features data from daily practice as well as external validation with data from clinical trials. We demonstrated that the LIPI score may represent an easy-to-use prognostic tool for patients with metastatic urothelial cancer who are eligible for ICIs. Furthermore, this tool that measures the immune-inflammatory status for each patient might be combined with other biological or genomic characteristics to develop more robust predictive models in the future.

## 5. Conclusions

The LIPI score appears to be an interesting prognostic biomarker irrespective of the type of systemic treatment in mUC. This score can be easily assessed with routine test access regardless of patient social, economic, and health insurance conditions, which may be limiting in some countries. Future prospective studies will be required to test the combination of the Bellmunt score and the LIPI score as an improved prognostic tool.

## Figures and Tables

**Figure 1 cancers-15-01066-f001:**
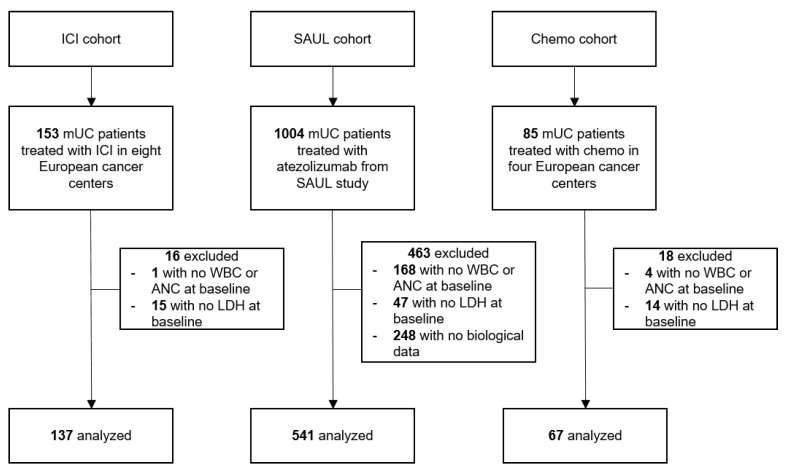
Flow diagram of the study population. ANC: absolute neutrophil count; Chemo: chemotherapy; ICI: Immune checkpoint inhibitor; LDH: lactate dehydrogenase; mUC: metastatic urothelial cancer; WBC: white blood cell.

**Figure 2 cancers-15-01066-f002:**
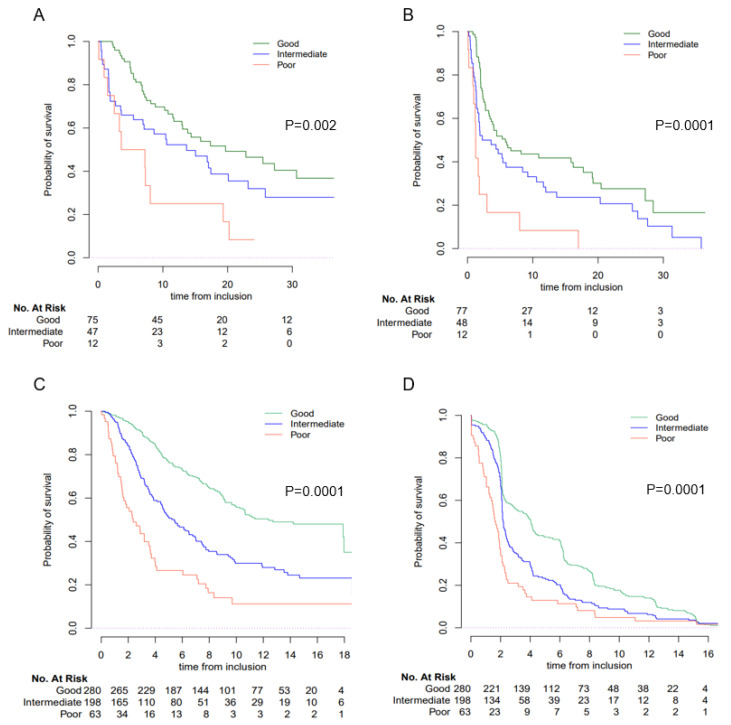
OS and PFS according to LIPI group in the ICI and SAUL cohorts. Overall survival (OS) and progression-free survival (PFS) according to the LIPI score. (**A**): OS in the ICI cohort, (**B**): PFS in the ICI cohort, (**C**): OS in the SAUL cohort, (**D**): PFS in the SAUL cohort.

**Figure 3 cancers-15-01066-f003:**
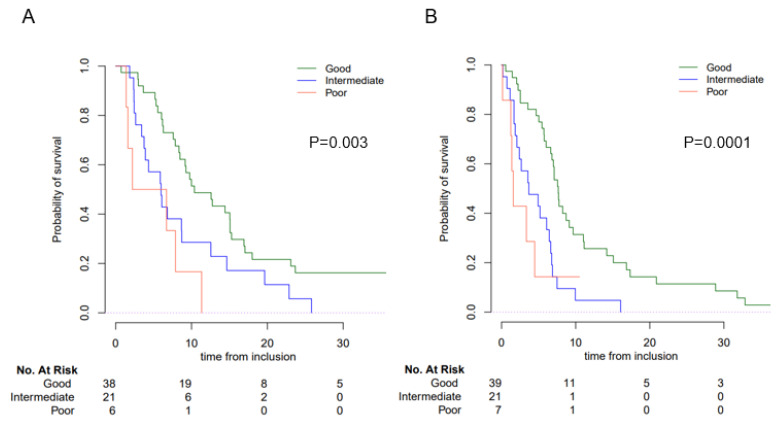
OS and PFS according to LIPI group in the Chemo cohort. Overall survival (OS) and progression-free survival (PFS) according to the LIPI score. (**A**) OS; (**B**) PFS.

**Figure 4 cancers-15-01066-f004:**
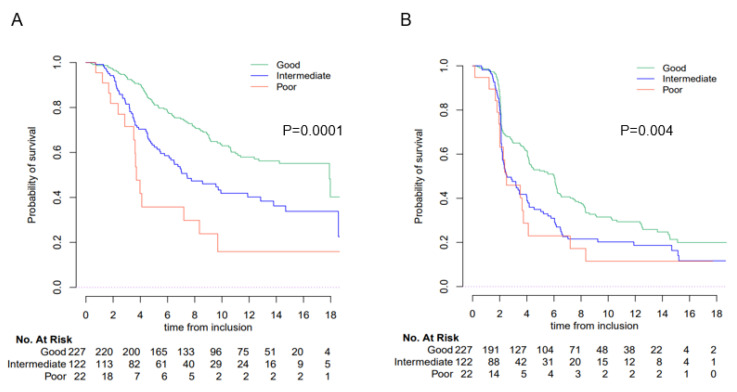
OS and PFS according to LIPI group in the SAUL cohort population with a Bellmunt risk factor of 0 or 1. Overall survival (OS) and progression-free survival (PFS) according to the LIPI score in the SAUL cohort population with a Bellmunt risk factor of 0 or 1. (**A**) OS; (**B**) PFS.

**Table 1 cancers-15-01066-t001:** Baseline patient and disease characteristics for the ICI, SAUL and Chemo cohorts. ANC: absolute neutrophil count; Chemo: chemotherapy; dl: deciliter; dNLR: derived neutrophils/(leukocytes minus neutrophils) ratio; ECOG: Eastern Cooperative Oncology Group; g: grams; ICI: immune checkpoint inhibitor; LDH: lactate dehydrogenase; LIPI: lung immune prognostic index; mo: months; n: number; NA: not available; PD-L1: programmed death-ligand 1; ULN: upper limit of normal.

Characteristics	ICI Cohort *N* = 137	SAUL Cohort *N* = 541	Chemo Cohort *N* = 67
**Gender** (n, %)			
Male	112 (82)	408 (75)	52 (78)
**Age at diagnosis** (yrs) (median, range)	68 [58;74]	67 [60;74]	68 [60;74]
**Smoking history** (n, %)			
>10 PYs	95 (69)	361 (67)	37 (55)
<10 PYs	31 (23)	180 (33)	19 (28)
Unknown	11 (8)	0 (0)	11 (17)
**Primary tumor** (n, %)			
Bladder	109 (80)	412 (76)	46 (69)
Upper tract	25 (18)	111(21)	21 (31)
Urethra	3 (2)	5 (1)	0 (0)
Other	0 (0)	13 (2)	0 (0)
**Histology** (n, %)			
Pure urothelial or mixed histology	128 (93)	519 (96)	64 (96)
Non-urothelial histology			
- Bellini collecting duct	0 (0)	5 (0)	0 (0)
- Glandular	1 (0)	3 (0)	0 (0)
- Neuroendocrine	2 (0)	4 (0)	2 (0)
- Squamous	6 (1)	8 (0)	1 (0)
Unknown	0 (0)	2 (0)	0 (0)
**Molecular FGFR alteration** (n, %)			
Yes	15 (11)		3 (4)
No	22 (16)	NA	8 (12)
Unknown	100 (73)		56 (84)
**PD-L1 status** (n, %)			
Positive	18 (13)	150 (28)	0 (0)
Negative	34 (25)	351 (65)	8 (12)
Unknown	85 (62)	40 (7)	59 (88)
**Type of prior treatment** (n, %)			
Platinum-based therapy	105 (77)	527 (97)	63 (94)
Gemcitabine	2 (1)	12 (2)	2 (3)
Vinflunine	7 (5)	1 (0)	7 (10)
Taxane	30 (22)	0 (0)	15 (22)
Other	8 (6)	3 (1)	7 (10)
**Pretreatment performance status (ECOG)** (n, %)			
0–1	106 (77)	489 (90)	46 (69)
≥2	29 (21)	52 (10)	21 (31)
Unknown	2 (2)	0 (0)	0 (0)
**Liver metastatic site** (n, %)	31 (23)	185 (34)	19 (28)
**No of prior treatment lines** (median, range)	1 (1–1)	1 (0–1)	2 (1–3)
**ICI treatment** (n, %)			
PD-L1 inhibitor	65 (47)	541 (100)	0 (0)
PD-1 inhibitor	72 (53)	0 (0)	0 (0)
**Circulating inflammatory markers** (median, range)			
Hemoglobin (g/dL)	12 (10.4; 13.1)	11.7 (10.5; 13.0)	11.9 (10.6; 13.2)
Leukocytes (Giga/L)	7.4 (5.9; 9.7)	7.5 (5.8; 9.5)	7.4 (5.8; 9.8)
ANC (Giga/L)	5.0 (3.7; 6.5)	5.0 (3.7; 7)	5.1 (3.4; 7.2)
Albumin (g/L)	40 (36; 42)	39 (35; 43)	36.5 (33; 41)
**LIPI score components** (n, %)			
**LDH > ULN**	33 (24)	169 (31)	16 (23)
**dNLR > 3**	39 (28)	155 (29)	19 (28)
**Follow-up** (mo) (median, 95% CI)	24 (19.8; 30.4)	13 (1.8; 13.2)	39 (33.7; NR)
**Radiological response** (n, %)			
Complete response	17 (12)	14 (3)	3 (5)
Partial response	27 (20)	48 (9)	24 (36)
Stable disease	26 (19)	175 (32)	21 (31)
Progressive disease	66 (48)	161 (30)	17 (25)
Non-valuable	1 (1)	143 (26)	2 (3)
**Progression-free survival** (mo) (median, range)	3.7 (2.7; 5.9)	2.2 (2.1; 2.3)	6.1 (4.7; 7.1)
**Overall survival** (mo) (median, range)	14.4 (10.8; 20.2)	8.7 (7.8; 9.9)	8.5 (6.3; 11.3)

**Table 2 cancers-15-01066-t002:** Multivariate analysis for OS and PFS in the ICI and SAUL cohorts; Multivariate analysis for OS in the Chemo cohort. Chemo: chemotherapy; CI: confidence interval; dl: deciliter; g: gram; HR: hazard ratio; ICI: Immune checkpoint inhibitor; LIPI: lung immune prognostic index; n: number; NA: not available; OS: overall survival; PFS: progression-free survival.

	ICI Cohort		SAUL Cohort		Chemo Cohort
Variables	HR (95% CI) PFS	HR (95% CI) OS	HR (95% CI) PFS	HR (95% CI) OS	HR (95% CI) OS
**Number of events/number of patients**	**100/130**	**74/127**	**476/489**	**276/489**	**45/49**
**Metastatic site**					
**Liver**					
Yes	1.55 (0.92; 2.61)	2.03 (1.15; 3.56)	1.68 (1.37; 2.05)	1.91 (1.48; 2.46)	NA
*p* value	0.096	0.013	<0.0001	<0.0001	
**Central nervous system**					
Yes	5.50 (2.01; 14.60)	NR	1.05 (0.50; 2.40)	0.88 (0.32; 2.39)	
*p* value	0.001		0.91	0.79	
**Pretreatment Performance status**					
≥2	2.53 (1.47; 4.33)	4.43 (2.48; 7.90)	2.73 (1.94; 3.83)	3.80 (2.58; 5.58)	0.83 (0.41; 1.66)
*p* value	0.001	<0.001	<0.0001	<0.0001	0.592
**Pretreatment**					
**Albumin**					
>35 g/L	1.00 (0.54; 1.85)	1.08 (0.53; 2.17)	0.81 (0.64; 1.02)	0.63 (0.48; 0.84)	0.22 (0.10; 0.45)
*p* value	0.99	0.84	0.08	0.002	<0.0001
**Pretreatment**					
**Hemoglobin**					
>10 g/dL	0.51 (0.28; 0.84)	0.50 (0.30; 1.98)	0.90 (0.68; 1.19)	0.93 (0.66; 1.30)	
*p* value	0.032	0.044	0.47	0.66	
**Pretreatment LIPI**					
Good LIPI	1 [reference]	1 [reference]	1 [reference]	1 [reference]	1 [reference]
Intermediate LIPI	1.68 (1.08; 2.61)	1.45 (0.86; 2.41)	1.24 (1.01; 1.51)	1.78 (1.35; 2.33)	1.25 (0.63; 2.33)
Poor LIPI	4.36 (2.04; 9.33)	2.69 (1.24; 5.84)	1.78 (1.29; 2.45)	2.89 (1.93; 4.32)	3.14 (1.07; 9.16)
*p* value	0.0003	0.035	0.001	<0.0001	0.05

**Table 3 cancers-15-01066-t003:** Comparison of outcomes with LIPI and Bellmunt scores in the SAUL cohort. HR: hazard ratio; CI: confidence interval; LIPI: lung immune prognostic index; NR: not reached; mRECIST: modified response evaluation criteria in solid tumors.

Characteristics	LIPI Score			Bellmunt Score			
	Good	Intermediate	Poor	0 Factor	1 Factor	2 Factors	3 Factors
**Patients** (n, %)	280 (52)	198 (36)	63 (12)	212 (26)	306 (37)	201 (24)	106 (13)
**Overall Survival**							
**Hazard ratio** (95% CI)	1 [ref]	1.78 (1.35; 2.33)	2.89 (1.93; 4.32)	1 [ref]	1.84 (1.37; 2.47)	4.47 (3.28; 6.09)	4.39 (2.70; 7.13)
**Median OS** (mo) (range)	12.4 (10.0; NR)	5.4 (4.5; 6.9)	2.4 (1.6; 3.7)	17.9 (14.5; NR)	8.6 (7.4; 11.9)	3.5 (3.0; 4.4)	2.0 (1.1; 4.7)
**Global log rank *p* value**	<0.0001			<0.0001			
**Progression-Free Survival according to mRECIST criteria**							
**Hazard ratio** (95% CI)	1 [ref]	1.24 (1.01; 1.51	1.77 (1.29; 2.45)	1 [ref]	1.25 (1.03; 1.51)	2.47 (1.97; 3.11)	3.05 (1.99; 4.66)
**Median PFS** (mo) (95% CI)	4.0 (3.1; 4.5)	2.2 (2.1; 2.4)	1.7 (1.4; 2.0)	5.3 (4.2; 6.2)	2.7 (2.8; 3.9)	2.0 (1.9; 2.1)	1.46 (1.1; 2.0)
**Global log rank *p* value**	0.001			0.001			
**Radiological response**							
**Disease control rate** (n, %)	137 (55)	59 (38)	8 (32)	125 (64)	125 (49)	44 (34)	6 (32)
***p* value**	0.001			<0.001			
**Objective response rate** (n, %)	38 (15)	17 (11)	5 (20)	41 (21)	44 (17)	12 (9)	1 (5)
***p* value**	0.334			0.024			
**Fast progressors** (n, %)	34 (12)	69 (35)	39 (62)	11 (5)	63 (21)	98 (49)	22 (61)
***p* value**	<0.001			<0.001			

## Data Availability

The data presented in this study are available on request from the corresponding author. The data are not publicly available due to ethical reasons.

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
