# Peer review of "Prognostic Value of the Lung Immune Prognosis Index Score for Patients Treated with Immune Checkpoint Inhibitors for Advanced or Metastatic Urinary Tract Carcinoma"

_cancers, 2023, doi:10.3390/cancers15041066_

Round 1

Reviewer 1 Report

In this study, Parent P. and colleagues evaluate, retrospectively, the prognostic power of the Lung Immune Prognosis Index (LIPI) in patients affected by advanced urothelial carcinoma previously treated with immune checkpoint inhibitors.

The main limitation of this study is its retrospective nature. Some imbalances of baseline characteristics of patients that have not been monitored or considered as well as differences in the follow-up length might limit the clinical utility of LIPI.

The validation of LIPI in a prospective study would be necessary to implement a new biomarker for metastatic urothelial carcinoma.

Minor comments:

In table 1, a line starting dNLR >3 (n,%)”,  there is a “<0.001” instead of 0(0).

Author Response

We would like to thank the reviewers for the careful and thorough reading of this manuscript and for the constructive comments and suggestions, which bolster our manuscript. All the comments were addressed and highlighted in the text.

Response to Reviewer 1 Comments

Comments and Suggestions for Authors

In this study, Parent P. and colleagues evaluate, retrospectively, the prognostic power of the Lung Immune Prognosis Index (LIPI) in patients affected by advanced urothelial carcinoma previously treated with immune checkpoint inhibitors.

The main limitation of this study is its retrospective nature. Some imbalances of baseline characteristics of patients that have not been monitored or considered as well as differences in the follow-up length might limit the clinical utility of LIPI.

The validation of LIPI in a prospective study would be necessary to implement a new biomarker for metastatic urothelial carcinoma.

We agree with the reviewer. We insisted on this general point in the Discussion part :

L316 We acknowledge several limitations of our study, including the retrospective design responsible for some imbalances in baseline patient characteristics and differences in follow-up, and its inherent selection bias and missing data.

Nevertheless, the strength of our study is that the LIPI was first studied on a retrospectively constructed cohort (LIPI Cohort) but was also validated on a homogeneous cohort of patients (SAUL Cohort) from a clinical trial whose data were collected prospectively. And in this cohort, the LIPI score is also prognostic.

Minor comments:

In table 1, a line starting “dNLR >3 (n,%)”,  there is a “<0.001” instead of 0(0).

We thank Reviewer 1 for his vigilance. We corrected this inconsistency.

Reviewer 2 Report

1_ Abstract: An abbreviation “pts” is not common in medical literature. It would be better not to use the abbreviation. 

2_ Please provide the definition of dNLR (formula) precisely in the abstract and the M&M section.

3_ A rationale of using dNLR rather than NLR: NLR is commonly used as an immuno-inflammatory biomarker in clinical studies on patients with malignancies. Please discuss a rationale by which dNLR is used as prognostic biomarker rather than NLR.

4_ The abstract may not reflect the design of this study, in which 3 cohorts including a control cohort of patients who received only chemo were used.

5_ Results: Please provide the number of outcome events to justify the number of variables included in multivariable analyses.

Author Response

We would like to thank the reviewers for the careful and thorough reading of this manuscript and for the constructive comments and suggestions, which bolster our manuscript. All the comments were addressed and highlighted in the text.

Response to Reviewer 2 Comments

1_ Abstract: An abbreviation “pts” is not common in medical literature. It would be better not to use the abbreviation. 

As recommended by Reviewer 2, we removed abbreviations.

2_ Please provide the definition of dNLR (formula) precisely in the abstract and the M&M section.

In the abstract, the definition of DNLR was given as follows : The LIPI score was based on 2 factors, derived neutrophils/(leukocytes minus neutrophils) ratio  (dNLR) > 3 and lactate dehydrogenase > upper limit of normal, and defined 3 prog groups.

In the M&M section, we added the definition as follow :

L121 The LIPI score was calculated as the original description. The LIPI score combines two variables : pretreatment LDH level,  and pretreatment dNLR derived neutrophils/(leukocytes minus neutrophils) ratio  defined 3 prognosis groups : (…)

3_ A rationale of using dNLR rather than NLR: NLR is commonly used as an immuno-inflammatory biomarker in clinical studies on patients with malignancies. Please discuss a rationale by which dNLR is used as prognostic biomarker rather than NLR.

We thank the reviewer for raising this point. In the litterature, the dNLR represents a well-correlated surrogate marker for the widely validated NLR. On the other hand, we have applied the original description of the LIPI score. In accordance with the suggestion of reviewer 2, in the introduction section, we supplemented and justified our choice for the dNLR biomarker with reference:

L74 The derived neutrophil to leukocyte ratio (dNLR) has been proposed as a potentially more relevant ratio, as it includes lymphocytes, but also monocytes and eosinophils thus better reflects the host immune status. In addition, dNLR has been used for NLR alternative in several clinical trials and independent significant association between high dNLR and poor OS as well as DFS outcomes was described in patients with malignancies  (20).

4_ The abstract may not reflect the design of this study, in which 3 cohorts including a control cohort of patients who received only chemo were used.

As suggested by the Reviewer 2, we included the chemotherapy cohort in the abstract

L38  A retrospective ICI cohort and a validation cohort (SAUL cohort) included respectively patients with mUC treated with ICI in 8 European centers (any line) and patients treated with atezolizumab in a second or further line. A chemotherapy-only cohort was also analyzed

 (…) L 46 Poor LIPI was associated with a poorer OS, with a hazard ratio (HR) =2.69 (95%CI; 1.24-5.84, p=0.035) for the ICI cohort and HR=2.89 (CI95%: 1.93-4.32, p<0.0001) for the SAUL cohort. Similar results were found in the chemo-cohort. 

5_ Results: Please provide the number of outcome events to justify the number of variables included in multivariable analyses.

 As recommanded by the Reviewer 2, we added the number of outcome events in table 2. We choose 1 variable for 8 events maximum.

Reviewer 3 Report

General comment

The manuscript entitled “Prognostic value of the Lung Immune Prognosis Index score for patients treated with immune checkpoint inhibitors for advanced or metastatic urinary tract carcinoma” aims to assess the role of LIPI in patients with metastatic urothelial cancer treated with ICI. Overall, the manuscript is fairly well written despite the particularly narrow topic of the paper. The main concerns regard the discussion that is in the initial part redundant with the results. The authors should expand this section by reporting similar and related studies (which however are few) and “defend” why LIPI could be an interesting score in those patients. Please also see: https://doi.org/10.3390/ijms23031133 and https://doi.org/10.3390/cancers14102545

Other suggestions in detail are reported followingly.

ABSTRACT

Avoid the use of abbreviations in the abstract.

Improve the clarity of results reported in the abstract.

INTRODUCTION

58: too simplistic, briefly expand.

66-67: add references.

91: It would be better to change with “the aim of this study”

PATIENTS AND METHODS

97: The number of patients should be reported in the results. Inclusion criteria could be more precise.

112: this cohort is more than a control group in this case.

126: Please define when you used chi-square or fisher's exact test.

143: add IRB number

RESULTS

Table 1: report p value among different groups involved.

Author Response

We would like to thank the reviewers for the careful and thorough reading of this manuscript and for the constructive comments and suggestions, which bolster our manuscript. All the comments were addressed and highlighted in the text.

Response to Reviewer 3 Comments

General comment

The manuscript entitled “Prognostic value of the Lung Immune Prognosis Index score for patients treated with immune checkpoint inhibitors for advanced or metastatic urinary tract carcinoma” aims to assess the role of LIPI in patients with metastatic urothelial cancer treated with ICI. Overall, the manuscript is fairly well written despite the particularly narrow topic of the paper. The main concerns regard the discussion that is in the initial part redundant with the results. The authors should expand this section by reporting similar and related studies (which however are few) and “defend” why LIPI could be an interesting score in those patients. Please also see: https://doi.org/10.3390/ijms23031133 and https://doi.org/10.3390/cancers14102545

We thank Reviewer 3 for his suggestions, we added a paragraph

L 292 Currently, no fully validated predictive biomarker is available to predict the prognosis and antitumor efficacy of ICI in patients with mUC. Programmed death ligand 1 expression, defective mismatch repair phenotype, and tumor mutational load have been studied as predictive biomarkers for management of urothelial cancer, but none is currently used in daily practice, mainly due to the lack of reliable reproducibility (29).

As an example, the lack of standardization of PD-L1 assessment across immunohistochemical assays, plus its dynamic expression make the results interpretation difficult (29). TMB seems to be correlated to treatment responses with ICI in different cancers and was investigated in mUC. The exploratory analyses of the phase II IMvigor210 trial showed that TMB was significantly higher in patients who responded to atezolizumab than in non-responders, however, the TMB was not able to perfectly predict the patients survival benefit (10).

Other biomarkers are emerging, such as circulating tumor DNA and microbiota, but these need to be validated in prospective clinical trials (30,31). High ctDNA levels have been shown to be correlate with more aggressive form of disease   and to be conversely correlated with overall survival in patients treated with durvalumab (32). In the IMvigor010 trial, patients with positive ctDNA had improved rates of disease-free survival and overall survival with adjuvant atezolizumab, compared to patients treated with a placebo, while no difference was reported between the treatment arms for ctDNA-negative patients.  Nevertheless, ctDNA level assessment is not implemented in clinical practice and the optimal predictive cut-off needs to be confirmed (33). In this sense, the standardized easy-to-used LIPI scoring algorithm, alone or combined with other biomarkers could erase their intrinsic weaknesses and better assess the probability of response to ICI in mUC, in order to avoid the exposure of patients to possible side effects when a minimal likelihood of the response is assumed.

Other suggestions in detail are reported followingly.

ABSTRACT

Avoid the use of abbreviations in the abstract.

As recommended by Reviewer, we removed abbreviations from the abstract except for the following : metastatic urothelial carcinoma (mUC) ; immune checkpoint inhibitors (ICI) ; The Lung immune prognostic index (LIPI)

Improve the clarity of results reported in the abstract.

As recommended by Reviewer we modified results section to improve the clarity

Results In the ICI and SAUL cohorts, 137 and 541 patients were respectively analyzed. In the ICI cohort, mPFS and mOS were 3.6 mo (95% CI; 2.6-6.0) and 13.8 mo (95% CI; 11.5-23.2) whereas in the SAUL cohort the mPFS and mOS were 2.2 mo (95% CI; 2.1-2.3) and 8.7mo (95% CI; 7.8-9.9). LIPI classified the population of these cohorts in good (56%; 52%), intermediate (35%; 36%) and poor (9%; 12%) prognotic groups (values for the ICI and SAUL cohorts respectively). Poor LIPI was associated with a poorer OS in both cohort, with a (hazard ratio (HR) for the ICI cohort =2.69 (95%CI; 1.24-5.84, p=0.035) for the ICI cohort and HR=2.89 for the SAUL cohort  (CI95%: 1.93-4.32, p<0.0001) for the SAUL cohort. Similar results were found in the chemo-cohort.  In patients with good prognostic according to the Bellmunt score, the LIPI identifies a subset of patients with poorer outcomes with mOS of 3.7 mo compared to the good and intermediate subgroups with mOS of 17.9 and 7.4 mo, respectively The LIPI score allows to identify different subgroups in patients with good prognosis according to the Bellmunt score criteria, with a subset of patients with poorer outcomes with mOS of 3.7 mo compared to the good and intermediate LIPI subgroups with mOS of 17.9 and 7.4 mo, respectively.

INTRODUCTION

58: too simplistic, briefly expand.

As Recommanded by the reviewer 3, we completed the sentence as follow : Platinum-based chemotherapy has been the mainstay of treatment in the metastatic setting, but outcomes are poor with an median overall survival (OS) of 12 to 15 months in patients eligible for first-line cisplatin-containing chemotherapy and around 9 months with patients “unfit” for cisplatin (2–4).

66-67: add references.

As Recommanded by the reviewer 3, we added references :

However, only a subset of patients benefits from ICI, highlighting the need of more accurate selection of patients who are more likely to respond : in Early pivotal trials in second-line and beyond therapy, the overall response rates (ORR) was assessed between 16% and 21 (5–8).

91  It would be better to change with “the aim of this study”

As Recommanded by the reviewer 3, we modified the sentence : . The hypothesis aim of this study was to identify

PATIENTS AND METHODS

97: The number of patients should be reported in the results. Inclusion criteria could be more precise.

The number of patients is already indicated in the results part, but we have removed it from methods section :

Three cohorts of patients with mUC were retrospectively analyzed. The first cohort (“ICI cohort”) included 153 patients…. Finally, the third cohort (“Chemo cohort”) included 67 patients treated only with chemotherapy in four European cancer centers between September 2011 to July 2019.

We added a sentence to precise inclusion criteria:

Inclusion criteria were patients treated from January 2015 to January 2019 for a mUC or unresecable urothelial carcinoma, in first line treatment or more (immunotherapy combinations were allowed but not combination of immunotherapy plus chemotherapy).

112: this cohort is more than a control group in this case.

We agree with the Reviewer 3.

126: Please define when you used chi-square or fisher's exact test.

The Cochrane recommends to use Fisher exact test when the number of patients is inferior to 20 (Cochran, WG. 1954. Somme methods for strengthening the common chi square tests", Biometrics, 10: 417-451).

We modified the sentence to : "(Fisher's exact test, when number of patients < 20)"

143: add IRB number

We added IRB number :

The study was approved by the Institutional Review Board (IRB) of Gustave Roussy (N°2020-01) and by the IRBs of all participating European centers

RESULTS

Table 1: report p value among different groups involved.

We cannot report the p-value between the three different cohorts because we did not compare them. The three cohorts were constructed differently:

First, we studied LIPI on a retrospectively constructed cohort (LIPI Cohort).

Then, we validated the LIPI score on a homogeneous cohort of patients (SAUL Cohort) from a clinical trial whose data were collected prospectively.

Finally, we tested the LIPI score in patients receiving chemotherapy only.

Round 2

Reviewer 1 Report

The authors have addressed my comments.

Reviewer 3 Report

No further corrections required.